# In Vitro Regeneration from Leaf Explants of *Helianthus verticillatus*, a Critically Endangered Sunflower

**DOI:** 10.3390/plants13020285

**Published:** 2024-01-18

**Authors:** Marzena Nowakowska, Zaklina Pavlovic, Marcin Nowicki, Sarah L. Boggess, Robert N. Trigiano

**Affiliations:** 1Department of Genetics, Breeding and Biotechnology of Vegetable Crops, The National Institute of Horticultural Research, 96-100 Skierniewice, Poland; 2Department of Entomology and Plant Pathology, Institute of Agriculture, University of Tennessee, Knoxville, TN 37996, USA; zpavlovi@alum.utk.edu (Z.P.); mnowicki@utk.edu (M.N.); sbogges1@vols.utk.edu (S.L.B.)

**Keywords:** Asteraceae, induction medium, morphogenetic response, whorled sunflower

## Abstract

*Helianthus verticillatus* (Asteraceae), a whorled sunflower, is a perennial species restricted to a few locations in the southeastern United States and is now considered endangered. Therefore, restoring and protecting *H. verticillatus* as a species is a priority. This study introduces a highly efficient in vitro adventitious plant regeneration system from leaf explants, utilizing five diverse specimens of *H. verticillatus*, each representing distinct genotypes with phenotypic variations in leaf and stem morphology. Key factors influencing in vitro morphogenesis, including genetic constitution, explant source, and plant growth regulators (PGRs), were identified. The study revealed a remarkably strong genotype-dependent impact on the regeneration efficiency of the investigated *H. verticillatus* genotypes, ranging from a lack of regeneration to highly effective regeneration. The selection of two genotypes with varying regeneration abilities provides valuable models for genetic analyses, offering insights into factors influencing the regeneration potential of this endangered species. Optimum adventitious shoot regeneration results were achieved using Murashige and Skoog basal media (MS) supplemented with 8.8 µM N6-benzyladenine (BA) and 1.08 µM α-naphthalene acetic acid (NAA). This combination yielded the highest adventitious shoot production. Subsequent successful rooting on ½ MS medium without PGRs further solidified the efficiency of the developed protocol. Regenerated plantlets, demonstrating robust shoots and roots, were successfully acclimatized to greenhouse conditions with a 95% survival rate. The protocol developed in this study is the first such report for this endangered species and is expected to contribute to future genetic manipulation and modification studies.

## 1. Introduction

Crop wild relatives serve as invaluable repositories of genetic diversity, offering opportunities to enhance the yield, quality, and adaptability of cultivated crops [1]. Within the Asteraceae, *Helianthus*, with more than 50 species, plays an important role in improving cultivated sunflower, *H. annuus* L., by providing genetic variability for traits like pathogen resistance, seed oil quality, and environmental stress tolerance [2,3].

*Helianthus verticillatus* Small, commonly known as the whorled sunflower, stands out as a perennial species within the *Helianthus* genus, characterized by its unique features. Designated as federally endangered in 2014 [4], *H. verticillatus* is restricted to a few locations in southeastern United States, underscoring its rarity [5,6,7]. Its appealing inflorescences, adorned with multiple showy yellow ray flowers, position it as a potential ornamental plant and a crucial resource for numerous insect pollinators [8,9]. Additionally, *H. verticillatus* seeds are rich in linoleic acid and low in saturated fatty acids, rendering them valuable for improving cultivated sunflower species [10].

Despite its significance, *H. verticillatus* encounters substantial challenges in propagation, attributed to its self-incompatibility, limited seed availability, and inefficiencies associated with traditional methods like rhizome digging and stem cuttings [6,7,8]. Seed-derived plants exhibit unique genotypes, but the preservation of distinct parental characteristics through ordinary seed or sexual reproduction practices is compromised [8,11]. The limited availability of seeds and the heterogeneous nature of seedlings pose constraints on propagation, despite the plant’s local invasiveness via rhizomes. Although rhizomes and stem cuttings are considered as alternative methods for propagation [8,12,13,14], they come with inherent limitations in large-scale production. The challenges encompass insufficient material supply, restricted seasonal propagation, and exposure of plant materials to various unfavorable factors, such as pests, pathogens, and drought [8,15,16,17,18].

Addressing these challenges, in vitro techniques emerge as promising solutions for the conservation and multiplication of endemic, rare, and endangered species, including *H. verticillatus*. Leveraging advantages like rapidity, space efficiency, and reduced time duration, in vitro methods, including axillary bud proliferation, have been identified as valuable for the conservation and multiplication of plant populations characterized by a small-size, low seed output, and/or viability [19,20,21,22,23,24,25,26,27]. Efforts in the *Helianthus* tissue culture have mainly focused on cultivated sunflowers (*H. annuus*) and *H. tuberosus* [28,29,30,31,32,33,34,35,36,37,38,39,40,41,42,43]. Limited data exist for other species, with only a few publications available [44,45,46,47,48,49,50]. Wild species, in some instances, show a higher response to tissue culture techniques than cultivated sunflowers. However, successful shoot regeneration is influenced by factors such as induction media composition, explant type, and specific plant genotype. Some *Helianthus* species, notably, resist adventive shoot induction, emphasizing the need for tailored regeneration protocols [36,47,51,52,53].

A notable knowledge gap exists concerning *H. verticillatus*, specifically regarding the adventitious shoot regeneration from leaf explants. While previous work from our group successfully established a protocol for generating genetically uniform plants from nodal meristems [21], our objective in this study was to develop a highly efficient in vitro regeneration protocol for *H. verticillatus* using leaf explants. This novel focus aimed to contribute to the future genetic improvement, including genetic transformation and gene function analysis, of this endangered species.

We hypothesized that *H. verticillatus*, similarly to other wild *Helianthus* species, may exhibit a high in vitro regeneration potential from leaves, with the regeneration capacity influenced by culture conditions. Additionally, we sought to investigate whether the genetically heterogeneous nature of *H. verticillatus* would impact the efficiency of regeneration, marking the first exploration of this genetic factor in wild sunflower species within the context of leaf regeneration. The protocol development was involved the following steps: shoot regeneration, rooting, and acclimatization. To the best of our knowledge, this is the first report of regeneration from leaves for this endangered species. The newly established protocol may offer a means for a broad range of research, including genetic transformation and analyses of gene functions. Furthermore, upon the validation of the genetic fidelity of regenerants in future studies, it may play a pivotal role not only in the in vitro multiplication of *H. verticillatus* but also as a viable method for conserving and restoring populations of this critically endangered plant species.

## 2. Results and Discussion

### 2.1. Optimization of Induction Medium for Shoot Induction from Leaf Explants of Helianthus verticillatus

Plant growth regulators (PGRs) are typically essential for inducing the morphogenic response in plants [30,53,54,55]. Specifically, cytokinin alone or a high cytokinin-to-low auxin ratio is essential for calli and adventitious shoot induction in *Helianthus* species [28,29,30,31,33,40,42,43,45,46,47,48,49,50]. However, these findings have been primarily limited to only several *Helianthus* species, and the in vitro regeneration of *H. verticillatus* from leaf explants has remained unexplored until now. To identify the optimum conditions for shoot regeneration from leaf explants in this species, several protocols available for other *Helianthus* species [40,45,47,49,50] were tested. Our preliminary findings demonstrated that the regeneration of shoots from leaf explants of *H. verticillatus* was influenced by the combination of PGRs in the induction medium, including their type and concentration, genotype, and the interactions between these variables (Table 1 and Appendix A). A distinction between direct and indirect adventitious shoot generation was not made as it would require extensive histological investigation and was beyond the main scope of the study.

None of the explants cultured in the medium without PGRs (MS0) exhibited any signs of morphogenetic responses, and after three weeks of cultivation, the explants became necrotic. Although the MS5 medium supplemented with zeatin as the sole PGR induced abundant calli formation, BA alone (MS6, MS7, MS8) led to substantial expansion and thickening of explants with scanty calli in a very few cases. However, contrary to the results observed for *H. smithii* [50] and *H. tuberosum* [40], none of the PGRs used alone induced shoot formation (Table 1). Adventitious shoot development was only promoted when both auxin (NAA) and cytokinin (BA) were present in the induction media (Table 1). These results corroborated general findings that a high cytokinin-to-low auxin ratio, along with the determination of the appropriate concentrations of these PGRs, is necessary for the simultaneous induction of calli and shoots in various plants, including some *Helianthus* species [44,45,47,48,49].

The combination of cytokinin in high amounts (2.22 µM of BA) and auxin in low amounts (0.27 µM of NAA) was effective for calli formation and its further dedifferentiation of somatic tissues in six wild sunflower species [47]. However, when this protocol was followed in our study, only calli production was observed in *H. verticillatus* plants. This finding might be related, but not limited, to the auxin level in the induction medium (MS4), which may not have been sufficient to induce shoot production in *H. verticillatus*. The observation that shoot regeneration was induced in the MS3 medium [45], which had the same concentration of BA (2.22 µM) as in the MS4 medium but with a higher NAA concentration (2.68 µM in MS3 vs. 0.27 µM in MS4), may further support this hypothesis (Table 1).

Shoot formation was observed in two (HV05, HV10) out of three genotypes tested in Experiment 1. Moreover, the regeneration efficiency was influenced by the type of induction medium, albeit in a genotype-dependent manner. Although no significant differences were observed, it is worth noting that when considering shoot regeneration potential, explants from HV05 exhibited a relatively more favorable response on the MS1 medium, whereas HV10 responded more favorably to the MS3 medium (Table 1).

The primary objective of this study was to find a relatively simple PGR regime that could readily promote adventitious regeneration from leaf explants in *H. verticillatus*. We considered treatments that failed within 12 weeks as unsuitable, and their regeneration potential was not explored further. Given that shoot production in *H. verticillatus* is genotype-dependent and influenced by the induction medium, the following two distinct approaches were selected for subsequent investigation: (1) with a higher cytokinin level than the auxin level (MS1, MS1CH) and (2) with an equal cytokinin level to auxin (MS3) (Table 1).

### 2.2. Effect of Genotype, Explant Source, and Medium on In Vitro Regeneration of H. verticillatus

To investigate factors that impacted adventitious shoot regeneration, in vitro and in vivo leaf explants of five *H. verticillatus* genotypes were cultured on the induction media described above. Five to seven days after cultivation, leaf explants enlarged and produced calli at the wound sites, as well as on both the upper and lower explants’ surfaces. Subsequently, after 21 days of cultivation, calli development was observed in a range from 94.6% to 100% of the leaf explants, with no differences either due to genotype, the source of explants, or the specific induction medium used (Appendix A). These findings indicate that none of the examined variables played a significant role in calli induction. It is noteworthy that during this phase, the initiation of shoot formation was sporadic, occurring only rarely. Notably, some of the cultured explants developed roots instead of shoots. Following an additional 21 days of cultivation, utilizing either a fresh medium with a consistent composition (MS1, MS1CH) or the appropriate regenerative medium (KR-R in the case of MS3), the morphogenic response exhibited a greater degree of diversification. This variation ranged from direct shoot bud formation on the explant to indirect differentiation of shoots on the calli formed, whereas some explants remained undifferentiated after calli formation (Figure 1a–j).

A three-factor analysis of variance was conducted to gain deeper insights into the factors influencing both the number of shoots produced per explant and the number of explants producing shoot induction. The results revealed that these variables were significantly impacted by the following two major factors: genotype and the source of the explants (*p* < 0.001; Table 2). Notably, the composition of the induction medium did not have a significant influence on the regeneration efficiency. Similarly, the triple interaction ‘genotype × plant source × induction medium’ was not significant for either of the investigated traits. However, significant disparities in the interactions among various factors were demonstrated, highlighting the complex nature of the shoot induction process. Specifically, the interactions involving ‘genotype × plant source’ (*p* < 0.05), ‘genotype × induction medium’ (*p* < 0.001), and ‘plant source × induction medium’ (*p* < 0.05) showed significant effects on the frequency of explants showing shoot induction. Regarding the number of shoots produced per explant, significant differences were identified for two specific interactions: ‘genotype × plant source’ (*p* < 0.001) and ‘genotype × induction medium’ (*p* < 0.01).

The genotype was identified as a significant factor influencing regeneration frequency (*p* < 0.001; Table 2). The significance of genotype interaction with both the type of culture medium used and the source of explants on regeneration efficiency was also observed (Table 2, Figure 2a,b and Figure 3a,b). Among the genotypes tested, HV13 showed a notably high level of in vitro regeneration response, with the highest frequency of the explant responding, as well as the highest number of shoots per explant, regardless of the induction media (Figure 2a,b) and the explants source (Figure 3a,b). Depending on the induction medium, the plant regeneration efficiency varied from 57.2% to 65.3% (Figure 2a), whereas the number of shoot buds induced per explant varied from 3.0 to 4.3 (Figure 2b) in the HV013 genotype. Contrastingly, HV04 demonstrated recalcitrance to regeneration; it only formed calli without any organogenic or embryogenic responses in leaf-derived calli. Three other genotypes (HV05, HV10, HV18) responded with significantly lower regeneration efficiency when compared to HV13 (Figure 2a,b and Figure 3a,b; Appendix A), regardless of the constitution of the induction medium.

Our observations align with previous studies, which suggested that in the vitro regeneration response to tissue culture is genetically determined and is similar to many other plant species, including *H. annuus* [28,29,30,31,33,41], *Echinacea purpurea* [56], *Chrysanthemum* [57], and *Hagenia abyssinica* [58], among others. Substantial variations in the morphogenic response among different genotypes may be attributed to distinct mechanisms that regulate endogenous PGR metabolism and/or content, particularly cytokinin levels during the induction period, which could affect the responsiveness to the exogenously applied PGRs [59].

The selection of two genotypes (HV13 and HV04) that exhibited varying regeneration abilities in this study provided valuable models for further genetic analysis to elucidate the factors underlying the regeneration potential. Considering the genotype-dependent predisposition for regeneration from leaf explants, conducting detailed genetic analyses on highly morphogenetic plants would be instrumental in understanding the diverse regeneration potential of *H. verticillatus*.

The regeneration response of leaf explants was also influenced by the leaf explant source (*p* < 0.001; Table 2) but in a manner dependent on the genotype. The interaction between the plant source and induction medium, however, was found to be non-significant and did not exert any appreciable influence on the regeneration efficiency. Generally, in comparison to the in vivo leaf explants, the in vitro explants exhibited a more favorable response and showed a higher frequency of shoot induction when they were cultured on the same induction medium (Figure 3a,b; Appendix A). Nevertheless, in the majority of cases, these differences were not significant, except for HV13. Depending on the genotype, the plant regeneration efficiency varied from 21.5% to 71.9% for the in vitro explants and from 10.2% to 52.3% for the in vivo explants (Figure 3a,b). The number of shoot buds induced per explant varied from 0.61 to 4.96 for the in vitro explants and from 0.18 to 1.83 for the in vivo explants in the genotypes used in the study (Figure 3a,b).

Comparable findings have been reported for *Sapium sebiferum* [60], *Jatropha curcas* [61,62], and *Paulownia tomentosa* [63], among other species. The differences in the competence for the shoot production of explants in vitro and in vivo may be due to the differences in their physiological developmental states, as well as the levels of endogenous PGRs present in both sources of explants [61].

All induction media used in this study enabled the regeneration of all tested genotypes from leaf explants, except HV04. Out of the three tested variables, the type of induction medium showed no impact on the regeneration efficiency (Table 2). In general, among the three different media examined, the induction media supplemented with 8.88 µM BA and 1.08 µM NAA (MS1, MS1CH) appeared to be more favorable for the morphological response of leaf explants compared to the medium supplemented with 2.20 µM BA and 2.68 µM NAA (MS3). However, the observed differences in terms of the number of shoots per cultured explant and the frequency of explant response among the tested induction media were not significant. Of particular interest was the comparison of media MS1 (8.8 µM BA, 1.08 µM NAA) and MS1CH (BA 8.8 µM, NAA 1.08 µM, CH 500 mg L^−1^). Yordanov et al. [49] found that CH (casein hydrolysate) combined with a higher concentration of cytokinin (8.8 µM BA) and a lower concentration of auxin (1.08 µM NAA) had a strong positive effect on the regeneration efficiency. In contrast to that study [49], no significant differences were observed between the medium supplemented with and without CH in our study.

On all induction media studied, most of the shoots originated directly from somatic leaf tissue or through indirect organogenesis, with an intervening callus phase. Somatic embryogenesis was not observed. These findings are not entirely consistent with previous observations for the plant regeneration of the interspecific hybrid *H. eggertii* × *H. annuus* on the MS1CH medium [49] and *H. smithii* [50], where both organogenesis and somatic embryogenesis were also noted. Notably, the PGR composition of the MS3 medium, which induced somatic embryo formation on leaf explants of *H. giganteus* [45], also resulted in shoot production on leaf explants of *H. verticillatus* in our study.

### 2.3. Shoot Elongation and Multiplication

Regardless of the genotype or the induction media employed, emerging shoots sometimes exhibited symptoms of vitrification or had an atypical appearance, characterized by a relatively large number of small shoots closely packed together, which resembled a crown-like structure. The occurrence of hyperhydricity during plant tissue culture has been related to the induction medium used, especially the type and concentration of cytokinins, in conjunction with a high endogenous secretion of PGRs in the donor plant explants [64]. Importantly, vitrified shoots were recovered and grew normally after transferring to a PGR-free medium, resulting in sufficiently developed plantlets. Loss of hyperhydricity after changing the properties of the medium was reported in other studies [65,66], in which unviable shoots were recovered, resulting in normal growth and development. A similar tendency was also observed during plant regeneration from leaf explants in *H. smithii* [50]. Otherwise, the recovery of shoots from plants with an atypical appearance was often successful, but in contrast to vitrified shoots, it was more difficult and required additional (up to four) transfers to obtain normally developed plantlets.

During the experiment, an interesting phenomenon was observed. Regardless of the induction medium and the donor genotype, the plantlets spontaneously propagated themselves by adventitious shoot formation on the basal part of the stem and the roots. A similar tendency was observed in *H. maximiliani* when the leaf explants were used for induction of somatic embryogenesis [48]. This mass propagation through adventitious shoots affected the regeneration efficiency, highlighting the substantial potential of *H. verticillatus* for regeneration in vitro.

### 2.4. Rooting and Acclimatization

Regenerated shoots were used for rooting experiments. Following our prior study on the micropropagation of *H. verticillatus* plants from auxiliary buds [21], the induction of roots with auxins was not necessary as the vast majority of shoots (from 88 up to 100%; Table 3) successfully developed roots within four weeks after placement on auxin-free ½ MS medium.

Furthermore, the roots exhibited a well-developed structure with abundant secondary branching (Figure 4a,b). Similarly, some authors have reported that rooting of various plant species, including *H. maximiliani*, can be easily achieved in a medium with no PGRs [48,67,68]. For comparison, shoots of meristematic origin of hybrid progenies involving four wild *Helianthus* species (*H. decapetalus*, *H. giganteus*, *H. mollis*, *H. strumosus*) rooted at an average frequency of 46 to 65% when subcultured on an auxin-free regeneration medium [31]. The spontaneous rooting in the absence of exogenous auxins can be attributed to the availability of endogenous auxins in the shoot apex, with these being transported downwards to create an auxin gradient which is required for root induction [69,70]. The spontaneous rooting of in vitro-regenerated plantlets represents an added advantage of our regeneration protocol as it eliminates the need for a separate auxin-containing medium for root induction. This is particularly important given that reports for other in vitro propagated *Helianthus* species indicated that the addition of auxin was necessary to achieve satisfactory rooting efficiency, often leading to challenges in the process [34,47].

The subsequent critical step involved the acclimatization of the in vitro-obtained plantlets to ambient growing conditions. *In vitro*-regenerated, rooted plantlets of *H. verticillatus* were transferred to a soilless mixture and acclimatized to greenhouse environmental conditions. Consistent with our previous finding [21], *H. verticillatus* plants showed no specific requirements for acclimatization. Of 100 plantlets transferred to the greenhouse (25 plantlets per genotype), 96% survived and produced healthy new growth under greenhouse conditions. Furthermore, no differences in hardening were observed for the different genotypes studied. This aligns with the observation that wild species typically display relatively high survival rates, as evidenced in *H. giganteus* [45], *H. smithii* [50], and *H. maximiliani* [48]. The plantlets produced in this study appeared psychologically healthy and morphologically similar to the donor plants (Figure 4c,d). As a result, the protocol established in this study not only ensures a rapid and cost-effective rooting process but also guarantees a high survival rate for this important species.

## 3. Materials and Methods

### 3.1. Plant Materials

Five *H. verticillatus* plants with phenotypic variation in leaf and stem morphology were selected from the collection of the University of Tennessee, Knoxville, TN, USA, (UT)as the source of explants for all tissue culture experiments. These plants were initially obtained from Madison Co., TN, USA, and Floyd Co., GA, USA, before *H. verticillatus* was classified as an endangered species [4].

### 3.2. Assessment of the Genetic Distance among the Donor Plants

Total genomic DNA was extracted from fresh leaves of five greenhouse-grown donor plants.

Leaf samples were homogenized in micro-centrifuge tubes using zirconia beads (BioSpec Products Inc., Bartlesville, OK, USA) using a Bead Mill 24 (Fisher Scientific, Waltham, MA, USA). DNA extraction was performed using a protocol of the E.Z.N.A. Plant DNA kit (Omega Bio-tek, Norcross, GA, USA) according to the manufacturer’s instructions. The concentration of DNA was measured using a NanoDrop ND-1000 spectrophotometer (NanoDrop Technologies, Wilmington, DE, USA) and the DNA was stored at −20 °C. A panel of 15 microsatellites, selected based on a previous report [71,72] and our transcriptome experiments (PRJNA778959), was utilized for this study (Appendix A). DNA amplification was performed in 10 μL reactions with 4 ng genomic DNA and 0.25 μM of each primer, following the recommended protocol for AccuStart II PCR SuperMix (Quantabio, Beverly, MA, USA). Reactions were performed using the following touchdown-PCR conditions: 95 °C for 3 min, followed by 10 cycles of 94 °C for 30 s, 65 °C lowering 1 °C per cycle to a final 55 °C for 30 s, and then 72 °C for 45 s, another 30 cycles of 94 °C for 30 s, 55 °C for 30 s, 72 °C for 45 s, and a final elongation step at 72 °C for 20 min [21]. PCR products were separated using the QIAxcel Capillary Electrophoresis System (QIAGEN, Valencia, CA, USA) and sized with 25 to 500 base pair (bp) size markers and an internal 15/600 bp alignment marker [73]. Raw allele length data were then converted into discrete allelic classes using Flexibin [74]. The resulting data set was used for all further analyses. The matrix of genetic distances was calculated in R version 3.6.1 using RStudio version 1.2.5019, and package *poppr* version 2.8.3 [75]. Bruno’s genetic distance method was used for this analysis [76]. Bootstrap support values for each split in the dendrogram were calculated over 1000 permutations of the dataset. Genotyping using 15 polymorphic markers allowed us to classify these five specimens of *H. verticillatus* into distinct genotypes (Appendix A).

### 3.3. Media and Culture Conditions

Except for the rooting media, all media used were based on full-strength MS basal medium [77] supplemented with 3% (*w*/*v*) sucrose (Thermo Fisher Scientific, Pittsburgh, PA, USA) and 0.75% (*w*/*v*) phytoagar (PhytoTechnology Laboratories, Shawnee Mission, KS, USA). Plant growth regulators (PGRs) were added at various concentrations, and the pH was adjusted to 5.8 using 1 M NaOH, before autoclaving at 121 °C for 20 min. The rooting medium contained half-strength salt MS medium (½-MS), 0.75% agar, and 1% sucrose, with no PGRs.

### 3.4. Experimental Design

Two series of experiments were conducted to establish an efficient in vitro regeneration protocol. The first experiment aimed to identify the optimum combination of PGRs for shoot regeneration in *H. verticillatus*. The second experiment assessed the regeneration efficiency of different *H. verticillatus* genotypes, both in vivo and in vitro propagated plants, using the best-performing protocols from the initial set of experiments.

#### 3.4.1. Experiment 1: Optimization of Induction Medium for Regeneration

Leaves from three in vivo propagated genotypes of *H. verticillatus* were utilized as the source of explants. The explants were cultured on several variants of MS medium (Appendix A) containing various PGR combination(s) commonly employed in published protocols for in vitro shoot regeneration of various *Helianthus* species [40,45,47,49,50]. Medium lacking PGRs served as the control.

The first three fully expanded young leaves at the top of approximately 2-month-old or younger stalks grown in the greenhouse were collected. Leaves were surface sterilized with 70% ethanol for 2 min, followed with 30% (*v*/*v*) bleach solution (sodium hypochlorite 6%; Clorox, Oakland, CA, USA) containing two drops of Triton X-100 for 15 min, and finally were washed five times with sterile double-distilled H_2_O. Leaves were cut transversely across the midrib into small pieces and placed in 90 mm diameter plastic Petri dishes with the adaxial surface down on the induction media. Following the specifications provided by the authors of the protocols used, the dishes were incubated at 22 °C under an 18 h photoperiod supplied by cool-white, fluorescent lamps, except those containing the MS3 medium, which were kept in darkness. Explants showing signs of regeneration events were transferred to the appropriate medium under the culture conditions described in the respective protocol. The experiments were conducted over 12 weeks.

#### 3.4.2. Experiment 2: Influence of Explant Source, Genotype, and Induction Medium on Shoot Regeneration

To compare the morphogenetic capacity of in vitro explants with their in vivo counterparts of five genotypes of *H. verticillatus*, fully expanded leaves were obtained directly from in vivo plants (as described previously). Additionally, leaves were obtained from the micropropageted plantlets, which were previously generated from nodal explants cultured on MS medium [21]. Based on the results of the first experiment, three treatments were chosen to study the regeneration efficiency of *H. verticillatus*.

As the primary objective of the initial Experiment 1 was to identify appropriate combinations of PGRs for further research, modifications were subsequently incorporated into individual protocols to enhance regeneration efficiency based on the results. The first protocol was modified by Yordanow et al. [49]. Leaf segments were incubated on MS1 (8.88 µM BA, 1.08 µM NAA) or MS1CH (8.88 µM BA, 1.08 µM NAA, 500 mg × L^−1^ casein hydrolysate (CH)) medium at 22 °C with an 18 h photoperiod with a dark preculture period of 5 days for 6 weeks with subculturing at three-week intervals. In their original protocol [49], explants after three weeks of culture on the initial medium (MS1, MS1CH) were transferred to an MSM medium (0.88 µM BA, 250 mg × L^−1^ CH) for further development. However, explants/calli placed on this medium turned brown after one or two weeks of culture, and explants were subsequently kept on MS1 or MS1CH medium for the next three weeks. After 6 weeks, explants were transferred to MSM medium and refreshed every three weeks over 12 weeks. Additionally, a slightly modified protocol described by Krasnyanski et al. [45] for indirect somatic embryogenesis in *H. giganteus* was followed. Explants were incubated on the induction medium MS3 (2.2 µM BA and 2.7 µM NAA) for three weeks at 22 °C in the dark. Then, explants were transferred to the regeneration medium KR-R (8.88 µM BA, 0.05 µM NAA) [45] and incubated under the same temperature in the light. The KR-R medium was refreshed every three weeks for 12 weeks.

### 3.5. Shoot Elongation and Multiplication

When regenerating shoots had reached 1 to 2 cm in length, they were excised and transferred to Magenta GA7™ vessels (Magenta Corporation, Chicago, IL, USA) containing 50 mL of ½ MS without PGRs for elongation and spontaneous rooting. Shoots that did not form roots and/or showed poor growth were transferred repetitively to fresh ½ MS salt medium at four-week intervals until roots emerged. For the acclimatization stage, in vitro rooted plantlets were washed with sterile, distilled water to remove adhering agar; then, they were transplanted into plastic pots filled with Promix BX Mycorrhizae (Premier Horticulture Inc., Quakertown, PA, USA) and grown under greenhouse conditions. The tops of the pots were covered with transparent plastic for one week to maintain high humidity. Each genotype was represented by 25 plantlets. The survival rate of the plantlets was evaluated eight weeks after transplanting.

### 3.6. Data Analysis

The treatments were performed with four (1st experiment) or eight (2nd experiment) replicates and eight explants in each replicate. Cultures were observed daily for any response. The frequency of calli formation (defined as the ratio of the number of explants that formed calli tissue to their total number, expressed as a percentage) was calculated. Plant regeneration frequency was expressed as the percentage of plated explants that regenerated shoots (mean ± SD) and the number of shoots produced per explant plated (mean ± SD).

Tissue culture data were analyzed using analysis of variance (ANOVA) in factorial one-way, two-way, or three-way fashions, when applicable, using R version 3.6.1 in RStudio version 1.1.456, with the packages: *car* version 3.0-5 and *agricolae* version 1.3-1 [78]. Subsequently, Tukey’s Honestly Significant Differences (HSD) at α = 0.05 were calculated using the same software. Standard errors and averages were calculated using MS Excel 2019 and R.

## 4. Conclusions

This study has, for the first time, demonstrated the feasibility of achieving high-frequency *in vitro* adventitious plant regeneration from *H. verticillatus* using leaf explants. We identified a suitable shoot induction media and PGR for the in vitro plant regeneration *H. verticillatus*. In vitro induction media, PGRs, and explant genotype played an important role, either independently or by their pairwise interaction, in the callus formation and shoot organogenesis of *H. verticillatus*. However, *H. verticillatus* generated roots within 4 weeks of culture in half strength of MS media which was sufficient for the in vivo plant establishment. Our study will hold important implications for the future in vitro plantlet regeneration in *H. verticillatus*. Further studies on the molecular and physiological aspects can identify the regulatory mechanisms of the in vitro organogenesis of *H. verticillatus*.

## Figures and Tables

**Figure 1 plants-13-00285-f001:**
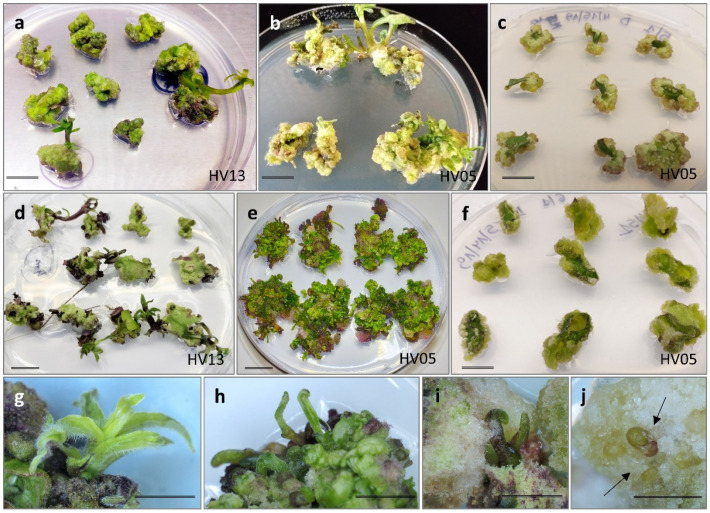
In vitro response from leaf explants of *Helianthus verticillatus* after six weeks on MS medium with 8.88 µM BA and 1.08 µM NAA (MS1) in 90 mm diameter Petri dishes: large and fast-growing shoots developed predominantly from meristematic nodules from in vitro (**a**) and in vivo (**d**) explants; clusters of shoot buds predominantly developed from calli formed from in vitro (**b**) and in vivo (**e**) explants; calli remained undifferentiated (**c**,**f**); vigorous and well-developed shoots (**g**,**h**), often covered by a more or less abundant calli (**i**,**j**). Bars in all figures = approximately 1.0 cm.

**Figure 2 plants-13-00285-f002:**
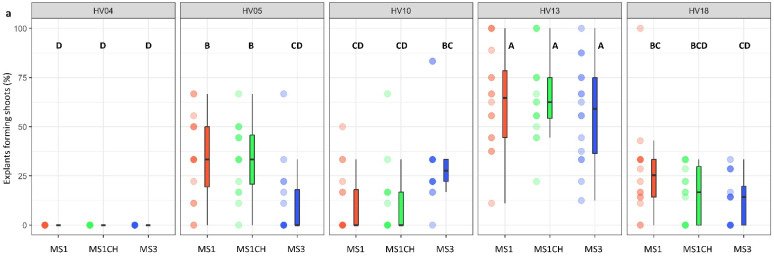
Effect of ‘genotype × induction medium’ interaction on the frequency of explants forming shoots (**a**) and the number of shoots per leaf explant (**b**) in five different genotypes of *Helianthus verticillatus*. Raw data are presented as a series of stacked bee swarms in colors representing three induction media, juxtaposed with the respective boxplots (median is marked in black; boxes represent the interquartile range; whiskers extend to cover the rest of the data in each group), with 192 explants represented each treatment. MS medium supplemented with 8.88 μM BA and 1.08 μM NAA (MS1); 8.88 μM BA, 1.08 μM NAA, and 500 mg × L^−1^ CH (MS1CH); 2.20 μM BA and 2.68 μM NAA (MS3). Due to lack of significance for the triple interaction ‘genotype × plant source × induction medium’ on plant regeneration frequency or the number of explants producing shoots, ‘genotype × induction medium’ pairwise interaction was investigated separately using two-way ANOVAs. Capital letters above each data stack represent grouping according to the Tukey test post-two-way-ANOVAs (‘genotype × induction medium’) at α = 0.05 (Appendix A). HSD (Honestly Significant Differences) for explants forming shoots = 21.74%, number of shoots per explant = 1.54.

**Figure 3 plants-13-00285-f003:**
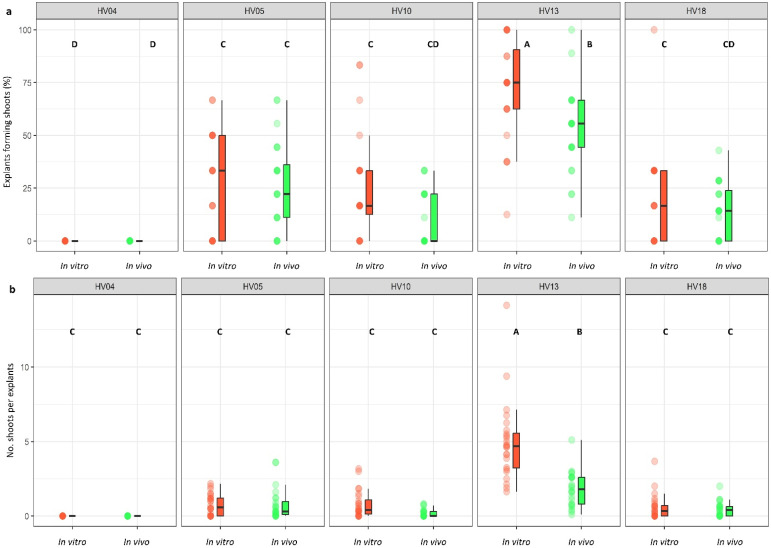
Effect of ‘genotype × plant source’ interaction on the frequency of explants forming shoots (**a**) and the number of shoots per leaf explant (**b**) in five different genotypes of *Helianthus verticillatus*. Raw data are presented as a series of stacked bee swarms in colors representing in vitro and in vivo explants, juxtaposed with the respective boxplots (median is marked in black; boxes represent the interquartile range; whiskers extend to cover the rest of the data in each group), with 128 explants represented each treatment. MS medium was supplemented with the following PGRs: 8.88 μM BA and 1.08 μM NAA (MS1); 8.88 μM BA, 1.08 μM NAA, and 500 mg × L^−1^ CH (MS1CH); 2.20 μM BA and 2.68 μM NAA (MS3). Due to lack of significance for the triple interaction ‘genotype × plant source × induction medium’ on plant regeneration frequency and the number of explants producing shoots, ‘genotype × plant source’ pairwise interaction was investigated separately using two-way ANOVAs. Capital letters above each data stack represent grouping according to the Tukey test post-two-way-ANOVAs (‘genotype × plant source’) at α = 0.05 (Appendix A). HSD (Honestly Significant Differences) for explants forming shoots = 16.94%, number of shoots per explant = 1.0.

**Figure 4 plants-13-00285-f004:**
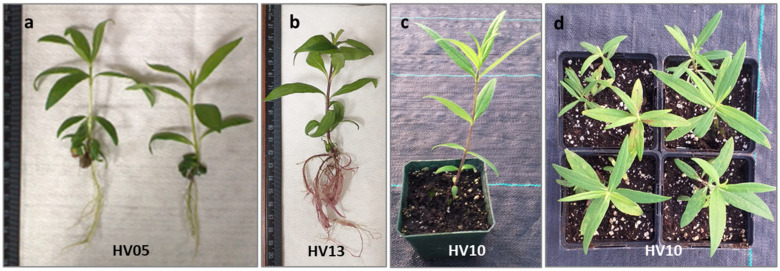
Regeneration of *Helianthus verticillatus*: shoot with well-developed leaves shoots showed healthy growth and successful rooting (**a**,**b**) after 4 weeks culturing on an auxin-free ½ MS; plantlet acclimatization in Promix BX Mycorrhizae under greenhouse conditions in square plastic pots (H: 9 cm, W: 8.5 cm) resulted in survived plantlets which produced healthy new aboveground vegetative growth 8 weeks after transplanting from in vitro conditions (**c**) and the subsequent vegetative season (propagated by rhizomes) (**d**).

**Table 1 plants-13-00285-t001:** The effects of different plant growth regulator (PGR) combinations on in vitro response of leaf explants from three *Helianthus verticillatus* genotypes in the first experiment optimizing induction medium.

Induction Medium	PGRs (μM)	Genotype	Explants Forming Callus (%)	Explants Forming Shoots (%)	No. of Shoots/Explant
MS0	-	HV04	0 c	0 d	0 c
		HV05	0 c	0 d	0 c
		HV10	0 c	0 d	0 c
MS1	BA (8.88) NAA (1.08)	HV04	100 a	0 d	0 c
		HV05	100 a	50.0 ± 14.4 a	1.43 ± 0.78 a
		HV10	100 a	5.6 ± 11.1 d	0.20 ± 0.4 bc
MS1CH	BA (8.88) NAA (1.08) CH (500 mg × L^−1^)	HV04	100 a	0 d	0 c
		HV05	97.2 ± 5.6 a	22.2 ± 9.1 bc	0.43 ± 0.26 bc
		HV10	100 a	5.6 ± 6.4 d	0.08 ± 0.09 bc
MS2	BA (4.44) NAA (0.54)	HV04	100 a	0 d	0 c
		HV05	100 a	11.1 ± 9.1 cd	0.23 ± 0.26 bc
		HV10	100 a	7.2 ± 14.3 d	0.08 ± 0.15 bc
MS3	BA (2.20) NAA (2.68)	HV04	100 a	0 d	0 c
		HV05	100 a	33.3 ± 9.1 b	0.63 ± 0.47 b
		HV10	100 a	25.0 ± 5.6 bc	0.45 ± 0.23 bc
MS4	BA (2.20) NAA (0.27)	HV04	100 a	0 d	0 c
		HV05	100 a	0 d	0 c
		HV10	100 a	0 d	0 c
MS5	ZEA^RIB^ (2.82)	HV04	100 a	0 d	0 c
		HV05	100 a	0 d	0 c
		HV10	96.4 ± 7.2 a	0 d	0 c
MS6	BA (2.22)	HV04	0 c	0 d	0 c
		HV05	0 c	0 d	0 c
		HV10	0 c	0 d	0 c
MS7	BA (4.44)	HV04	5.6 ± 11.1 c	0 d	0 c
		HV05	22.2 ± 18.1 b	0 d	0 c
		HV10	0 c	0 d	0 c
MS8	BA (8.88)	HV04	0 c	0 d	0 c
		HV05	0 c	0 d	0 c
		HV10	0 c	0 d	0 c

For each genotype, data are the average (±SD) of 28 explants. Letters represent grouping according to Tukey tests post two-way ANOVAs (‘induction medium’ × ‘genotype’) at α = 0.05 (Appendix A). HSD (Honestly Significant Differences) for: explants forming calli = 11.55%, explants forming shoots = 14.86%, and number of shoots per explant = 0.56.

**Table 2 plants-13-00285-t002:** Three-way ANOVA test to evaluate the significance of *Helianthus verticillatus* genotype, explant source, and induction medium on the frequency of calli induction, plant regeneration frequency, and the number of explants producing shoots.

Tested Parameters	Variation Source	Sum of Squares	F Value	*p* Value [Pr (>F)]
Explants forming calli (%)	Genotype (A)	63.3	1.2	3.14 × 10^−1 ns^
	Explant Source (B)	93.8	7.1	8.00 × 10^−3^ **
	Induction Medium (C)	39.1	1.5	2.31 × 10^−1 ns^
	A × B	30.1	0.6	6.86 × 10^−1 ns^
	A × C	43.6	0.4	9.13 × 10^−1 ns^
	B × C	76.2	2.9	5.80 × 10^−2 ns^
	A × B × C	86.1	0.8	5.92 × 10^−1 ns^
Explants forming shoots (%)	Genotype (A)	100,694.0	88.1	<2.20 × 10^−16^ ***
	Explant Source (B)	4358.0	15.3	1.27 × 10^−4^ ***
	Induction Medium (C)	941.0	1.6	1.95 × 10^−1 ns^
	A × B	3624.0	3.2	1.48 × 10^−2^ *
	A × C	12,242.0	5.4	3.90 × 10^−6^ ***
	B × C	1989.0	3.5	3.26 × 10^−2^ *
	A × B × C	2390.0	1.1	4.03 × 10^−1 ns^
Number of shoots per explants	Genotype (A)	348.7	82.3	<2.20 × 10^−16^ ***
	Explant Source (B)	36.9	34.8	1.43 × 10^−8^ ***
	Induction Medium (C)	3.7	1.8	1.75 × 10^−1 ns^
	A × B	85.1	20.1	5.09 × 10^−14^ ***
	A × C	27.2	3.2	1.85 × 10^−3^ **
	B × C	2.9	1.4	2.60 × 10^−1 ns^
	A × B × C	14.7	1.7	9.22 × 10^−2 ns^

* Significant at *p* < 0.05; ** significant at *p* < 0.01; *** significant at *p* < 0.001; ^ns^ not significant. The triple interaction ‘genotype × plant source × induction medium’ was not significant for either of the traits. Subsequently, two-way ANOVAs were conducted to test all possible pairwise interactions due to the significant interactions between various factors (Figure 2a,b and Figure 3a,b; Appendix A).

**Table 3 plants-13-00285-t003:** Frequency of rooting of *Helianthus verticillatus* shoots regenerated from leaf explants depending on explant source, induction media, and genotype.

Explant Source	Induction Medium *	Rooting Percentage			
		Genotype			
		HV05	HV10	HV13	HV18
In vivo	MS1	100.0	100.0	98.6	100.0
	MS1CH	100.0	100.0	98.8	100.0
	MS3	100.0	100.0	89.8	100.0
In vitro	MS1	100.0	100.0	87.5	100.0
	MS1CH	89.3	100.0	88.6	100.0
	MS3	100.0	100.0	87.7	100.0

Data are the average raw counts obtained for shoots induced from the leaf explants scored after 4 weeks in culture on an auxin-free ½ MS. The number of rooted shoots varied depending on the genotype, inducing medium, and explant origin. * MS medium supplemented with the following PGRs: 8.88 μM BA and 1.08 μM NAA (MS1); 8.88 μM BA, 1.08 μM NAA, and 500 mg × L^−1^ CH (MS1CH); 2.20 μM BA and 2.68 μM NAA (MS3).

## Data Availability

The original contributions presented in the study are included in the article/Appendix A, further inquiries can be directed to the corresponding author/s.

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
