# Peer review of "In Vitro Regeneration from Leaf Explants of *Helianthus verticillatus*, a Critically Endangered Sunflower"

_plants, 2024, doi:10.3390/plants13020285_

Round 1
Reviewer 1 Report (Previous Reviewer 2)
Comments and Suggestions for Authors
I appreciate the effort and time that the authors put into addressing all of my comments. I'm happy with the final version and have no additional critical comments.
Author Response
We would like to thank you for your constructive feedback and the time you dedicated to reviewing our work. We are pleased to hear that you are satisfied with the final version and have no additional critical comments.
Reviewer 2 Report (Previous Reviewer 1)
Comments and Suggestions for Authors
As can be seen from the Abstract, Introduction and Conclusion, the procedure of in vitro regeneration from leaves is intended as (i) a novel and suitable regeneration protocol for in vitro multiplication of H. verticillatus, as (ii) a procedure for future genetic manipulations and modifications, and also as (ii) a possible method for biodiversity conservation and preservation of genetic resources of this endangered species. While the first two reasons are well documented and supported by the results, for the third one the manuscript lacks supporting data from experiments as well as arguments from relevant literary sources.
It is a generally accepted phenomenon that the in vitro indirect regeneration from somatic tissues brings an increased risk of genetic and epigenetic changes, which is not appropriate in cases where we want to multiply and conserve genetic resources of a certain species, and to preserve biodiversity corresponding to the situation on the site. It would therefore be correct in this case if the authors also added an experimentally verified claim that the regenerants were genetically identical to the initial explants. I therefore recommend to mention the possible risks and not to connect this method with the conservation of genetic resources of H. verticillatus (see lines 30, 65-66, and 503-504), as its suitability for this purpose has not been verified.
The second serious shortcoming (from my point of view) is often subjective and unbalanced literature review concerning the advantages of this method compared to other multiplication methods (see lines 48-58). For this purpose, the authors used three literary sources (#6-8), which do not sufficiently document the disadvantages of the other methods, and in some cases they even sound in their favour. In several cases, literary sources are missing (e.g. lines 64-66 "Additionally, other in vitro methods, especially the adventive formation of plants from somatic, non-meristematic tissues, play a pivotal role in conservation management").
Although the manuscript contains novel, interesting and beneficial results, the above shortcomings must be resolved.
Author Response
We would like to thank you for your valuable feedback on our manuscript. We appreciate the detailed analysis and observations you provided. Please find the detailed responses below and the corresponding revisions highlighted in yellow in the re-submitted files.
To address the description of our in vitro leaf regeneration procedure for H. verticillatus, we took into consideration your suggestions regarding the three main objectives of the procedure, and we rephrased our manuscript to address your concerns, especially those related to the biodiversity conservation aspect. We acknowledge the lack of experimental data supporting clonal fidelity of regenerants. Concerning the risk of genetic and epigenetic changes associated with in vitro regeneration, we recognize the importance of this issue. However, organogenesis doesn’t necessarily guarantee non-clonal fidelity – it may or may not, with some changes being very subtle, whereas others (such as the lack of chlorophyll) are obvious. We had molecular data -- the SSR data (a similarity matrix based on 15 EST-SSRs indicated 100% identity in vitro regenerants to their respective donor plants) – obviously too small of a genetic sample to detect any changes. Perhaps GBS data could be useful for this purpose but it was beyond the scope of our study and this manuscript.
Regarding the second comment about the unbalanced literature review, we have incorporated additional sources to provide a more balanced perspective, discussing not only the advantages of in vitro methods but also the benefits of traditional vegetative methods of propagation. Recent research has explored the potential of in vitro cultures as a promising alternative or complementary approach to traditional methods, addressing their shortcomings and providing a means to overcome challenges associated with large-scale production, seasonality, and environmental factors.
In accordance with Academic Editor’s recommendation, we have removed lines 64-66 from the manuscript.
We trust that these revisions adequately address your concerns and enhance the overall quality of the manuscript. We appreciate your time and effort in reviewing our manuscript.
This manuscript is a resubmission of an earlier submission. The following is a list of the peer review reports and author responses from that submission.
Round 1
Reviewer 1 Report
Comments and Suggestions for Authors The manuscript is clearly and comprehensibly written, it brings interesting and valuable results concerning the in vitro multiplication of the endangered plant Helianthus verticillatus. The text has no major flaws, but the Introduction lacks information that this plant can be propagated well using rhizomes and seeds as well (as a starting information source, you can use e.g. https://ecos.fws.gov/ServCat/DownloadFile/233646 .) Therefore it is necessary to supplement this information in the Introduction and possibly slightly modify the justification why it is important to propagate this plant using in vitro techniques.
Author Response
The manuscript is clearly and comprehensibly written, it brings interesting and valuable results concerning the in vitro multiplication of the endangered plant Helianthus verticillatus. The text has no major flaws, but the Introduction lacks information that this plant can be propagated well using rhizomes and seeds as well (as a starting information source, you can use e.g. https://ecos.fws.gov/ServCat/DownloadFile/233646) Therefore it is necessary to supplement this information in the Introduction and possibly slightly modify the justification why it is important to propagate this plant using in vitro techniques.
We thank the reviewer for taking the time to review this manuscript and all comments on the manuscript. Please find the detailed responses below and the corresponding revisions highlighted in yellow in the re-submitted files.
We believe the introduction is sufficient to inform the readership of propagation methods. We have added the comment on seeds as presented in Trigiano et al. 2021. We would like to emphasize that H. verticillatus could be conserved by seeds, however, considering the self-incompatible character of this species, the seeds derived from cross-pollination show a high level of genetic variation. Indeed, the whorled sunflower is locally invasive via rhizomes, which may be dug and used to propagate the plant, but in our experience are not efficient. Therefore, adventitious regeneration from leaf explants is considered the most suitable way for possible subsequent studies: protoplast isolation and fusion, development of an efficient transformation system regeneration, in vitro mutagenesis, and transcriptome studies, among others.
Reviewer 2 Report
Comments and Suggestions for Authors
The work presented is very well explained, thorough and clear. The experiments are correct and so are the conclusions obtained. However, the main problem observed for its publication is the similarity between this work and the one previously published in this same journal (Plants 2020, 9(6), 712; https://doi.org/10.3390/plants9060712) by the same authors. This paper studies the regeneration of H. verticillatus using leaf explants versus Axillary Bud. This detracts from the novelty of the work and it should not be published in this journal.
The following comments are added to improve the work
Manor concerns:
1) Table 1 (page 3/17) gives the % callus formation, shoot formation and no. of shoots/ explant for 3 different genotypes, depending on the induction medium used, varying in the combination and/or concentration of hormones. However, I do not understand why only the data for 3 of the 5 genotypes used in the experiments are given. The table only shows the data for genotypes HV04, HV05 and HV10. Why don't they include the results for genotypes HV13 and HV18?
2) In line 136 on page 4/17 the authors state that the genotype HV05 showed better shoot formation response and no. of shoots/ explant when MS1 and MS1CH induction media were used (referring to table 1). However, if you look at Table 1, the results obtained for that genotype are better with the MS3 medium than for the MS1CH medium.
3) In Figure 1, on page 5/17, to which genotypes do the different explants shown belong? Were phenotypic differences observed in explants of different genotypes? I.e. were the callus more or less friable, greener or browner.
4) On lines 240-243 the authors state that compared to in vitro explants. In vivo explants showed a more favorable response to shoot induction, with higher shoot frequency when grown on the same induction medium as in vitro explants. However, just a few lines down, from 244-247, they indicate the opposite. They say that, depending on the genotype, regeneration efficiency varies from 21.5-71.9% in in vitro explants, in contrast to in vivo explants, which showed a efficiency of 10.2-52.3%.
The same for the number of shoots: for in vitro explants ranged from 0.61-4.96, while for in vivo explants ranged from 0.18-1.83. Is it not a contradiction to state that the efficiency is higher in in vivo explants, but show data of higher regeneration in in vitro explants?
Minor comments:
Page 1, line 17: in vitro in italics.
Page 13, line 436: Experiment 1 or 2.
Author Response
The work presented is very well explained, thorough, and clear. The experiments are correct and so are the conclusions obtained. However, the main problem observed for its publication is the similarity between this work and the one previously published in this same journal (Plants 2020, 9(6), 712; https://doi.org/10.3390/plants9060712) by the same authors. This paper studies the regeneration of H. verticillatus using leaf explants versus Axillary Bud. This detracts from the novelty of the work and it should not be published in this journal.
We thank the reviewer for taking the time to review this manuscript and all comments on the manuscript. Please find the detailed responses below and the corresponding revisions highlighted in yellow in the re-submitted files.
We disagree strongly with the evaluation of the referee that there is “too much similarity” between this report and (Plants 2020, 9(6), 712; https://doi.org/10.3390/plants9060712). All in vitro papers share common attributes. For example, media, PGRs, sometimes experimental designs, statistical methods, etc. This report primarily deals with adventitious regeneration, not axillary bud proliferation – two very distinctive processes and different uses. True, we did not make the distinction between direct (without a callus intermediate) and indirect (callus intermediate) – histological evaluation was completely beyond the study’s objective – regeneration from leaves.
Point 1: Table 1 (page 3/17) gives the % callus formation, shoot formation and no. of shoots/ explant for 3 different genotypes, depending on the induction medium used, varying in the combination and/or concentration of hormones. However, I do not understand why only the data for 3 of the 5 genotypes used in the experiments are given. The table only shows the data for genotypes HV04, HV05 and HV10. Why don't they include the results for genotypes HV13 and HV18?
We appreciate your attention to detail regarding Table 1, which presents the results from Experiment 1 (initial) aimed at determining optimal conditions for adventitious shoot induction, focusing on three specific genotypes. The rationale behind including data for only these genotypes was explained in the Materials and Methods section.
Point 2: In line 136 on page 4/17 the authors state that the genotype HV05 showed better shoot formation response and no. of shoots/ explant when MS1 and MS1CH induction media were used (referring to table 1). However, if you look at Table 1, the results obtained for that genotype are better with the MS3 medium than for the MS1CH medium.
Thank you for bringing this to our attention, and we appreciate the clarification on the specific observation. We acknowledge the oversight in our initial statement regarding genotype HV05 and its shoot formation response. The corrected information is now accurately reflected in the text.
Point 3: In Figure 1, on page 5/17, to which genotypes do the different explants shown belong? Were phenotypic differences observed in explants of different genotypes? I.e. were the callus more or less friable, greener or browner.
We appreciate the reviewer's attention to Figure 1, specifically inquiring about the genotypes corresponding to different explants and whether phenotypic differences were observed among them. The images refer to genotypes HV013 and HV05.
During our observations, we noted phenotypic variations in the callus, which exhibited differences in texture (ranging from friable to compact) and color (various shades of green, sometimes displaying reddish tint, especially where explants originated from plants with intense stem pigmentation). However, it is crucial to emphasize that these differences were not genetically determined, nor were they attributed to variations in the composition of the induction medium. The observed diversity was quite substantial even within a single plate, which could indicate the influence of other factors on callus diversity (such as the specific part of the leaf from which the explant originated).
Point 4: On lines 240-243 the authors state that compared to in vitro explants. In vivo explants showed a more favorable response to shoot induction, with higher shoot frequency when grown on the same induction medium as in vitro explants. However, just a few lines down, from 244-247, they indicate the opposite. They say that, depending on the genotype, regeneration efficiency varies from 21.5-71.9% in in vitro explants, in contrast to in vivo explants, which showed a efficiency of 10.2-52.3%. The same for the number of shoots: for in vitro explants ranged from 0.61-4.96, while for in vivo explants ranged from 0.18-1.83. Is it not a contradiction to state that the efficiency is higher in in vivo explants, but show data of higher regeneration in in vitro explants?
We appreciate the reviewer's keen observation regarding a discrepancy in the text. On lines 240-243, there was an error in stating that in vivo explants showed a more favorable response to shoot induction compared to in vitro explants. This was a mistake, and we have corrected the text accordingly.
Point 5: Page 1, line 17: in vitro in italics. Corrected.
Point 6: Page 13, line 436: Experiment 1 or 2. Corrected.
Reviewer 3 Report
Comments and Suggestions for Authors
There is nothing novel for advancing plant biotechnology/plant tissue culture in this study, although some practical data have been collected. The early events regarding in vitro morphogenesis have not been proved by histology, so it's not very clear about their regeneration pathway. The suggestion is to transfer it to plant tissue culture-specialized or horticulture journals. Please see details;
-This manuscript chiefly deals with a tissue culture study on sunflowers, which can be thought of as a routine job nowadays in the field of plant science. I believe there are too many tissue-culture labs that can do the same thing, I mean there is nothing novel in the study.
-Nothing new and can not make advances in plant biology or biotechnology. What does it add to the subject area compared with other published material? The same group has published similar reports that provided reliable regeneration protocols more efficient than this current work, see
1. Nowakowska M, Pavlović Ž, Nowicki M, Boggess SL, Trigiano RN. In Vitro Propagation of an Endangered Helianthus Verticillatus by Axillary Bud Proliferation. Plants (Basel). 2020 Jun 3;9(6):712. doi: 10.3390/plants9060712. PMID: 32503227; PMCID: PMC7356533.
2. Trigiano, R.N.; Boggess, S.L.; Wyman, C.R.; Hadziabdic, D.; Wilson, S. Propagation Methods for the Conservation and Preservation of the Endangered Whorled Sunflower (Helianthus verticillatus). Plants 2021, 10, 1565. https://doi.org/10.3390/plants10081565
-The authors should perform histology to confirm the regeneration pathway particular for the early stage of re-differentiation from the callus.
-Lack of the evidence to clarify the morphogenesis pathway.
-The references need to discuss the novelty when compared with their previous similar reports.
The tables and figures are lacks of scale bars in all photos.
Author Response
This manuscript primarily focuses on a tissue culture study of sunflowers, a routine undertaking in contemporary plant science. The study lacks novelty, and given the standard nature of the work, it is suggested that it be transferred to journals specializing in plant tissue culture or horticulture.
We thank the reviewer for taking the time to review this manuscript and all comments on the manuscript. Please find the detailed responses below and the corresponding revisions highlighted in yellow in the re-submitted files.
Point 1: The study, centered on sunflower tissue culture, is considered routine in the current landscape of plant science. The lack of novelty diminishes its contribution to the field. The research does not introduce anything new or advance plant biology or biotechnology. Its value in comparison to other published materials is unclear. The authors have previously released similar reports, such as the one referenced below, which provided more efficient regeneration protocols: Nowakowska M, Pavlović Ž, Nowicki M, Boggess SL, Trigiano RN. In Vitro Propagation of an Endangered Helianthus Verticillatus by Axillary Bud Proliferation. Plants (Basel). 2020 Jun 3;9(6):712. doi: 10.3390/plants9060712. PMID: 32503227; PMCID: PMC7356533.
Trigiano, R.N.; Boggess, S.L.; Wyman, C.R.; Hadziabdic, D.; Wilson, S. Propagation Methods for the Conservation and Preservation of the Endangered Whorled Sunflower (Helianthus verticillatus). Plants 2021, 10, 1565. https://doi.org/10.3390/plants10081565
We appreciate the reviewer's comments and concerns regarding the perceived lack of novelty in our study on sunflower tissue culture. While we acknowledge the existence of previous reports, such as those referenced in the review, we would like to emphasize the distinct focus and contributions of our current work. Our report is not similar to the other reports.
Trigiano, R.N.; Boggess, S.L.; Wyman, C.R.; Hadziabdic, D.; Wilson, S. Propagation methods for the conservation and preservation of the endangered whorled sunflower (Helianthus verticillatus). 2021, 10, 1565, doi:10.3390/plants10081565T deals with seed propagation and propagation via cuttings from existing plants and Nowakowska, M.; Pavlović, Ž.; Nowicki, M.; Boggess, S.L.; Trigiano, R.N. In vitro propagation of an endangered Helianthus verticillatus by axillary bud proliferation. Plants 2020, 9, 712, doi:10.3390/plants9060712 is about axillary bud proliferation. These reports are distinctly different than the present report.
Point 2: Histological analysis is recommended to validate the regeneration pathway, especially concerning the early stages of re-differentiation from the callus. Insufficient evidence is provided to elucidate the morphogenesis pathway.
Histological studies were completely beyond the scope of the study. We have made adjustments to the language that states we did NOT make the distinction between direct and indirect regeneration as for this study it was not important. Our primary goal was to be able to produce plants from somatic sources such as leaves. These protocols could be useful for future transformation, genetic, physiological, and morphological studies and therefore are valuable and very much different from our previous studies.
Point 3: The references should address the novelty of the current study in comparison with the authors' previous similar reports.
We have revised the manuscript to address and incorporate these comments.
Point 4: Tables and figures lack scale bars in all photos, which is a necessary improvement for clarity and precision.
We would like to explain that at this stage, unfortunately, we are unable to add scale bars to the images. We also believe that it could clutter the images and, more importantly, is not crucial to the main content of our article.
Round 2
Reviewer 2 Report
Comments and Suggestions for Authors
Thank you very much for the improvements made but if the article wants to be published, as I previously commented to the editor, the abstract and introduction should be modified as it is almost identical to that of the previously published paper.
Reviewer 3 Report
Comments and Suggestions for Authors
I remain my suggestion to reject it for publication because the novelty of this manuscript is quite low and the authors do not attempt to improve the quality accordingly.